# DNA Repair in Space and Time: Safeguarding the Genome with the Cohesin Complex

**DOI:** 10.3390/genes13020198

**Published:** 2022-01-22

**Authors:** Jamie Phipps, Karine Dubrana

**Affiliations:** UMR Stabilité Génétique Cellules Souches et Radiations, INSERM, iRCM/IBFJ CEA, Université de Paris and Université Paris-Saclay, F-92265 Fontenay-aux-Roses, France; jamie.phipps@cea.fr

**Keywords:** DNA repair, NHEJ, homologous recombination, chromatin, nuclear organization, chromatin dynamics, cohesin

## Abstract

DNA double-strand breaks (DSBs) are a deleterious form of DNA damage, which must be robustly addressed to ensure genome stability. Defective repair can result in chromosome loss, point mutations, loss of heterozygosity or chromosomal rearrangements, which could lead to oncogenesis or cell death. We explore the requirements for the successful repair of DNA DSBs by non-homologous end joining and homology-directed repair (HDR) mechanisms in relation to genome folding and dynamics. On the occurrence of a DSB, local and global chromatin composition and dynamics, as well as 3D genome organization and break localization within the nuclear space, influence how repair proceeds. The cohesin complex is increasingly implicated as a key regulator of the genome, influencing chromatin composition and dynamics, and crucially genome organization through folding chromosomes by an active loop extrusion mechanism, and maintaining sister chromatid cohesion. Here, we consider how this complex is now emerging as a key player in the DNA damage response, influencing repair pathway choice and efficiency.

## 1. Introduction: DNA Double-Strand Breaks Repair and Genome Stability

To ensure genome stability, DNA damage by both endogenous and extrinsic sources must be dealt with robustly. Without effective mechanisms to detect and repair assaults on the genome, diseases such as cancer can arise [1]. DNA double-strand breaks (DSBs) are particularly deleterious. If unrepaired, DSBs can result in chromosome loss and, if repaired incorrectly, can lead to point mutations, loss of heterozygosity and chromosomal rearrangements [2], all of which could lead to oncogenesis or cell death. 

In eukaryotes, including yeast and humans, DSBs are predominantly repaired by two mechanisms: non-homologous end joining (NHEJ), and homologous recombination (HR; Figure 1). NHEJ ligates two DSB ends in a homology-independent manner [3]. Although accurate when re-ligation takes place without DNA processing, NHEJ can lead to genome alteration by the loss or addition of nucleotides [4] or chromosomal translocations [5]. In contrast, HR uses an intact homologous donor sequence to reconstitute broken DNA. Typically, use of the homologous sister chromatid during HR results in faithful DNA DSB repair, although, if performed between alleles or heterologous sequences, transfer of mutation or loss of heterozygosity can occur. Although NHEJ and HR are the predominant pathways used for DNA DSB repair, other mechanisms are also observed; however, these are often less faithful. If NHEJ is compromised, repair by alternative end joining (a-EJ) pathways can take place. Repair by microhomology-mediated end joining (MMEJ) is dependent on the annealing of roughly 4–20 bp of microhomology close to both ends of the DSB, which are exposed after limited end resection, and generates small deletions [6]. Alternatively, longer direct homologous repeats that are unmasked by resection can be repaired by single-strand annealing (SSA), in a process that also sees the loss of the genomic sequence that once separated them [6].

Upon DSB, the first repair pathway engaged is NHEJ, which relies on the rapid recruitment of the KU heterodimer (along with DNA-PK in mammalian cells; Table 1) and the XRCC4–XLF–Ligase IV ligation complex [7]. If ligation fails, DNA resection is initiated at the break site by the Mre11/Sae2^CtIP^ complex, unmasking short 3′-single-stranded DNA (ssDNA) overhangs of 60–70 bp [8,9,10]. This limited resection may unmask short direct repeats, the annealing of which, followed by DNA synthesis by DNA polymerases (Pol θ in mammals, Polδ and Pol4 in yeast) mediate repair by MMEJ [6]. If resection proceeds further, mediated by the partially redundant nuclease activity of Dna2/Sgs1^BLM^ and Exo1 [10], longer 3′-ssDNA overhangs are generated that can engage in homology-directed repair (HDR). The 3′-ssDNA overhangs are rapidly stabilized by replication protein A (RPA), which in turn is replaced by the Rad51 recombinase via the Rad52 recombinase mediator. The resulting right-handed helical filament can invade the homologous donor DNA duplex, ultimately leading to DNA synthesis and the sealing of the DSB, followed by resolution of intermediate recombination structures. Long-range resection may also unmask longer direct repeats that can anneal in a Rad52-dependent manner to mediate repair by SSA. Resolution of SSA intermediates is achieved by the Rad1–Rad10 complex, which removes the 3′ non-homologous tail generated. This pathway does not require the invasion of a donor DNA duplex and is, therefore, Rad51 independent (for more details on the mechanisms see [11,12]). 

Repair pathway choice, thus, relies primarily on resection initiation, which is highly regulated at several levels. Notably, the stage of the cell cycle plays a key role, with HR favored in the S phase due to the stimulation of resection by cyclin-dependent kinases (CDKs). The local sequence context, chromatin composition and fiber dynamics, as well as the global nuclear architecture, also regulate repair pathway choice and repair completion. In this review, we explore the requirements for successful NHEJ and homology-directed repair (HDR), particularly, the local chromatin context of the broken DNA molecule, the movement dynamics of DSB ends, the global chromatin context that makes the donor sequence permissive to homology search and the influence of nuclear structures and localization within the nuclear space on DSB repair. We consider how this affects repair choice and efficiency, and throughout, we discuss how the cohesin complex modulates these aspects and is emerging as a key player in DNA repair.

## 2. Cohesin Structure and Loop Extrusion Activity in *Saccharomyces cerevisiae* and Humans

Cohesin is a multiprotein, ring-shaped complex, which was initially identified in budding yeast, and is conserved in almost all eukaryotes (Figure 2A). The complex was first described to hold sister chromatids together from S phase to anaphase, entrapping them to ensure equal division of chromosomes (Figure 2B; [13]). However, cohesin has increasingly been implicated in novel functions, including the 3D organization of chromatin by the formation of long-range intrachromatid loops (Figure 2C; [14,15]), likely by the extrusion of chromatin in a symmetrical manner [16,17]. 

In *S. cerevisiae*, cohesin consists of four core and essential subunits: the structural maintenance of chromosomes (SMC) proteins, Smc1 and Smc3, the kleisin Scc1 and the kleisin-associating Scc3 (Figure 2A; [18,19]). The following are the human orthologs: SMC1, SMC3, SCC1 and STAG1/STAG2, respectively (Figure 2A; Table 2). SMC proteins consist of “head” and “hinge” domains, separated by a long antiparallel coiled-coil arm. The head comprises the N and C terminal domains that, respectively, provide the A and B motifs of a Walker ATPase [20]. The hinge is generated where the coiled coil, which separates the two halves of the head domain, reverses direction. Smc1 and Smc3 heterodimerize through their hinge domains, as well as making contacts through their head domains, which are essential for ATPase activity [18,21]. The Scc1 subunit binds Smc3 at its N terminal and Smc1 at its C terminal, generating separate Smc and kleisin compartments when the ATPase heads are engaged, upon the binding of two ATP molecules. The Scc3 subunit binds to the central domain of Scc1, completing the cohesin complex [18,22].

Other proteins, such as Scc2/Scc4 (NIPBLA/NIPBLB–Mau2 in humans), Pds5 (PDS5A/PDS5B in humans) and Wpl1 (WAPL in humans) also bind to the complex, through the Scc1 recruitment platform (Figure 2A; Table 2; [23,24,25]). These dynamic interactions facilitate cohesin loading (Scc2/4) and dissociation (Wpl1) or, in the case of Pds5, have a dual role in the establishment and maintenance of sister chromatid cohesion, as well as dissociation through recruiting Wpl1 [26,27]. Cohesin is loaded onto chromosomes prior to the S phase by Scc2/4, which causes a conformational change in the cohesin complex. This opens the cohesin ring and allows it to embrace DNA, potentially through the hinge domains or the Smc3–Scc1 interface [27,28,29]. Once loaded, Smc3 acetylation by the acetyltransferase Eco1 (ESCO1 and ESCO2 in humans) stabilizes cohesin chromosome embracement by antagonizing Wpl1 [30,31]. At this point, a DNA-replication-coupled process leads to cohesin-dependent cohesion of sister chromatids [32,33]. Timely sister chromatid separation is regulated by Scc1 cleavage by a cysteine protease, separase (Esp1 in *S. cerevisiae*), during the anaphase [13,34].

Various conformations of the human cohesin complex have been identified by advanced microscopy techniques, which provide insight into how it facilitates both its sister chromatid cohesion and loop extrusion functions. These in vitro studies indicate that the ATPase SMC heads can be engaged, separated or juxtaposed, in a dynamic manner that is regulated by ATP binding (engaged) and hydrolysis (separated/juxtaposed) (Figure 2; reviewed in [19,29,35]). Engagement of the ATPase heads upon ATP binding confers a conformation in which the coiled-coil arms are separated, generating distinct SMC and kleisin compartments (Figure 2A). In the ATP-unbound state, ATPase heads can be separated or juxtaposed. When separated, the coiled-coil arms do not align, generating one open SMC–kleisin compartment (Figure 2D). In the juxtaposed state, the SMC coiled-coils align, generating a rod-shaped complex, in which only a juxtaposed kleisin compartment is present (Figure 2D). Alignment of the coiled coil is permissive to bending at an elbow region within the arms, which can bring the hinge domain into close contact with the SMC3 head domain (Figure 2D; [29]). 

Crucially, various DNA binding domains throughout the cohesin complex, as well as the loading partner NIPBL, have been shown to be essential for in vitro loop extrusion activity by human cohesin [29]. It appears that ATP- and DNA-binding-dependent conformation changes within the cohesin complex facilitate the passing over of the DNA molecule between DNA binding sites, although the full sequence and order of these events remains unclear, with multiple models being proposed [16,19,29]. The importance of DNA entrapment within the different compartments of the complex also remains unclear for the loop extrusion process. 

Loop extrusion by the budding yeast cohesin complex has not been formally demonstrated. However, the observation of cohesin-dependent loops, which expand when cohesin residency time is increased by Wpl1 depletion, argues in favor of loop extrusion [15,36]. Furthermore, in vitro studies have demonstrated the ability of budding yeast cohesin to bridge DNA molecules and compact DNA [37,38]. Unlike human cohesin, yeast cohesin forms molecular condensates upon interactions with DNA, leading to pronounced clustering [38,39]. Although the biological significance of this in vitro observation remains to be fully demonstrated, recent cryo-EM observations of budding yeast MRX, also an SMC family complex, revealed that it shares an ability to form large condensates, and crucially responsible protein motifs were identified [40]. Whether these motifs in Mre11 are conserved in cohesin and are relevant for their clustering activity remains to be determined.

The loop extrusion activity of the cohesin complex has revealed its importance for a broad range of DNA-related processes that go beyond its role in sister chromatid cohesion. These include regulation of gene transcription and, significantly, the DNA damage response in both yeast and mammals [41], due to its ability to shape the genome, and influence chromatin composition and nuclear architecture on multiple levels.

## 3. Chromosome Organization within the Nuclear Space and Cohesin Contribution

Eukaryotic genomes are organized at multiple levels, and ultimately exist in a highly folded state. The first level of chromatin folding consists of the periodic wrapping of the DNA double helix around a core of histone octamers to form nucleosomal chromatin fibers. These fibers are further organized into topologically associated domains (TADs), which have defined boundaries and exhibit increased local interactions within them and decreased interactions between them [35,42]. The mammalian genome is partitioned into a succession of TADs, which range in size from tens of kilobases to 1–2 Mb of DNA, whereas in yeast, smaller TAD-like structures have been described (50–100 kb in *S. pombe* and 5 kb in *S. cerevisiae*). Current models propose that cohesin forms TADs by loop extrusion between boundary proteins such as CTCF in mammals, or CARs (cohesin-associated regions) in yeast [15,36,43,44,45]. Cohesin also contributes to the higher-order organization of TADs, into TAD cliques, in which increased interactions are observed between distant TADs, in a constitutive or dynamic manner [46]. The contribution of cohesin to the individualization of chromosome domains imposes a constraint on the distance between sequences in the nucleus. This constraint could favor or disfavor contacts between DNA sequences during DNA repair and modulate both the DNA damage response and outcome, as supported by recent studies described below.

On a larger scale, chromatin is separated into different states with distinct characteristics, defined by specific histone variants, post-translational modifications (PTMs) and chromatin-binding proteins. Traditionally, two broad categories of chromatin states are distinguished, the transcriptionally active euchromatin and the densely packed and repressive heterochromatin, that overlap respectively with two compartments, A or B, defined by increased long-range interchromosomal interactions [47,48,49]. Cohesin is not required to form these compartments and rather appears to counteract their folding, as cohesin depletion results in an enhancement of A/B compartmentalization, as observed by increased contrast in Hi-C contact patterns [43,44,45,50,51]. Conversely, increasing loop formation by the depletion of WAPL or PDS5 strongly inhibits chromatin compartmentalization [45]. How cohesin opposes compartment formation remains to be defined experimentally. However, polymer simulations suggest that this could be achieved by cohesin-mediated loop extrusion [52].

Heterochromatin itself is subdivided into the ubiquitous constitutive heterochromatin, associated with highly repetitive sequences [53], and the more dynamic and often developmentally regulated facultative heterochromatin [54]. Heterochromatin can be associated with nuclear structures, including the nuclear lamina, forming lamina-associated domains, further organizing chromosomes within the nucleus [55]. An intriguing link exists between pericentromeric heterochromatin and cohesin in several organisms. In vertebrates, despite previous conflicting reports, a recent study demonstrated that haspin, the inhibitor of the cohesin-releasing factor WAPL, interacts with the heterochromatin protein HP1 in pericentromeric heterochromatin [56]. This interaction prevents premature dissociation of centromeric cohesin and ensures that cohesion is protected in pericentromeric heterochromatin at early stages of mitosis [56]. Haspin also cooperates with cohesin in interphase to ensure robust polycomb-dependent homeotic gene silencing in *Drosophila* [57]. In *S. pombe*, the Psc3 (Scc3 in budding yeast) cohesin subunit directly interacts with the heterochromatin protein Swi6, which ensures cohesin recruitment and cohesion establishment at centromeres but is also important to ensure the genomic integrity of the heterochromatic mating type locus [58]. Finally, cohesin is enriched in subtelomeric regions and is required for their transcriptional repression in both fission and budding yeast through mechanisms that remain to be deciphered [59,60]. How cohesin shapes these compacted regions and whether this influences gene expression or DNA repair remains an open question.

Beyond these substructures and compartments, chromosomes fold on themselves, defining chromosome territories with few interchromosomal interactions in mammals [48,61]. Additionally, homologous chromosomes are separated in the somatic cells of most diploid organisms [62,63,64,65] and are even more distant than expected in human cells, an organization that appears to be actively defined [66]. These characteristics are likely to be significant in disfavoring recombination events between distinct chromosomes, but this remains to be experimentally tested. In yeast, chromosome territories are less clear, but the spatial arrangement of chromosomes imposed by the tethering of the centromeres at one pole and the clustering of telomeres at the nuclear periphery [67,68] favors interactions between clustered sequences. Several studies have revealed a clear correlation between physical distance and recombination efficiency, with closest loci recombining with higher efficiency [69,70,71,72]. Beyond physical distance, other factors influence recombination efficiency. For example, in vivo studies have shown that limiting the rate of resection can increase recombination efficiency at some subtelomeric and intrachromosomal DSBs [71,72], demonstrating a relationship between the rate of resection and successful homology search (reviewed in [73]).

Although the genome is actively folded and ordered within the nucleus, this organization is not static, and movement of the chromatin fiber is observed to a similar extent in all organisms, with single loci exploring volumes with a radius of 0.5 to 1 µm [74]. In normal conditions, chromatin exhibits a subdiffusive motion, reflecting constrained movement. The first constraint on chromatin motion is linked to its polymeric nature and its higher-order folding. In addition, external factors such as crowding and viscoelastic properties of the environment, as well as interaction with nuclear substructures, in particular with the nuclear membrane, also impinge on motion (reviewed in [75,76]). Chromatin motion is an energy-dependent process that is reduced upon glucose starvation or the depletion of intracellular ATP [77,78,79]. Part of this dependency on ATP could be linked to ATP-dependent chromatin remodelers that have been shown to drive enhanced chromatin mobility (reviewed in [75,76]). In *S. cerevisiae*, the cell cycle stage also has a dramatic effect on motion, which is restrained during the S phase and is much more dynamic during G1. Reduced motion in the S phase is replication and cohesin dependent, as S phase inactivation of cohesin restores mobility to G1 levels [80,81]. This S-phase-specific effect has led to the proposal that the cohesion between sister chromatids restrains chromatin motion. However, chromatin mobility is constant throughout the interphase in mammals, with depletion of cohesin also increasing chromatin mobility [82,83]. This suggests a sister chromatid cohesion independent role for cohesin in influencing chromatin motion, which could rely on its ATP-dependent loop extrusion activity. While chromatin motion is regulated in a conserved manner, its significance for cellular processes is far from clear. Indeed, a number of studies have described changes in chromatin motion in response to DNA damage, while the relevance of these changes for DNA repair have not been fully defined.

## 4. Genome Folding and Chromatin Dynamics Modulate DNA Repair

The final 3D architecture of the genome, defined through the combined influence of its structure at multiple levels, as well as the nuclear structures to which it is associated, provides both structural and regulatory functions that modulate DNA repair. 

Breaks induced in different chromatin contexts lead to varied responses to DSBs, supporting a role of pre-established chromatin marks in DSB repair choice. Indeed, DSB repair pathway usage and efficiency in various chromatin environments has been addressed by employing genome-wide analysis of repair in euchromatic DSB sites [84] or the repair of specific heterochromatic sites [72,85,86,87,88,89,90,91,92,93]. The various forms of chromatin interfere with the recruitment of DSB repair proteins, thus contributing to DSB processing and DNA repair pathway choice. HR was shown to be the prevalent repair mechanism for endonuclease-induced DSB sites in transcriptionally active genes in human cell lines, while noncoding or silent euchromatic sequences exhibit a preference for NHEJ [84,94]. The H3K36me3 histone mark, typical of actively transcribed euchromatin, is thought to promote HR through the recruitment of the protein LEDGF, which mediates the recruitment of CtIP and, therefore, triggers ssDNA formation, Rad51 loading and HR initiation [84,95,96]. In parallel, the active chromatin mark H4K16-Ac, catalyzed by the TIP60 acetyltransferase, inhibits the binding of the anti-resection and pro-NHEJ factor 53BP1, thus favoring resection and HR commitment [97]. In contrast, H3K27me3-associated heterochromatin, or chromatin targeted to the repressive nuclear lamina, was shown to favor repair by NHEJ or alt-NHEJ through an undefined mechanism [89]. 

Paradoxically, the repair of DSBs in constitutive heterochromatic regions also appears to rely heavily on HR in different organisms [85,90,98,99], as observed in repeat-rich regions in G2 mouse cells [92,100] and in *Drosophila* pericentromeric heterochromatin [88]. This is partially due to the heterochromatin protein HP1, which recruits BRCA1 [101] to promote resection, as well as the recruitment of TIP60 by H3K9me3, which may promote decompaction of the DSB-flanking chromatin [102]. This decompaction is accompanied by the exclusion of the DSB to the periphery of the heterochromatin clusters, as observed in both *Drosophila* and mice [88,92,103]. In *Drosophila*, but not in mammals, exclusion from heterochromatin domains is followed by migration to the nuclear periphery, where HR takes place [88]. These studies support a model in which HR is actively repressed in heterochromatin domains. These relocation events, which isolate resected DSB from the bulk of heterochromatin, are proposed to help prevent recombination between the highly repetitive heterochromatic sequences, limiting sequence loss. Recent reports analyzing repair outcomes at unique genomic sites cleaved by meganucleases or CRISPR-Cas9 found no major change in the balance between NHEJ and HR when comparing heterochromatin and euchromatin [91,104]. However, observations in *Drosophila* suggest that heterochromatic repair might require specific DSB-induced chromatin modifications, involving a histone demethylase, to achieve the same NHEJ/HR balance seen in euchromatin [105]. The exact mechanism at work is still under investigation.

Lastly, a high-throughput study using CRISPR-Cas9 cleavage of a unique cassette inserted throughout the genome by a PiggyBac transposase optimized system has addressed the repair of DSB sites by NHEJ and MMEJ, depending on the chromatin context [106]. Although NHEJ is generally the most frequent repair pathway, how it is outcompeted by MMEJ varies depending on the chromatin context. Notably, the H3K27me3 heterochromatin mark favors MMEJ at the expense of NHEJ, suggesting it could promote resection initiation [106]. 

In *S. cerevisiae*, heterochromatin clearly modulates repair pathway choice through the control of resection at several levels [72,93]. The compacted chromatin structure modulates long-range resection through a still unknown mechanism [72]. In addition, Sir3, the mammalian HP1 functional ortholog, suppresses resection initiation through direct interaction and inhibition of the MRX^MRN^ activator Sae2^CtIP^. This in turn promotes NHEJ and protects heterochromatin from unscheduled HR [93]. Notably, although delayed by resection inhibition, HR repair is proficient in yeast heterochromatin. Limiting resection is of particular importance at subtelomeric DSBs as it avoids loss of chromosome end sequences and favors repair by conservative HR [72].

Beyond the chromatin context, an increasing number of studies have highlighted the contribution of higher-order chromatin structures, chromosome organization and interaction with nuclear substructures, such as the nuclear membrane, to DNA damage signaling and repair. As previously stated, the successive layers of genome folding—from TADs, TAD cliques, compartments and whole chromosome territories to chromosome positioning within the nucleus—each constrain contact between genomic sequences. These structures likely regulate HR, which is highly dependent on contact between the damaged DNA and the homologous template [73]. Furthermore, they could elicit the illegitimate rejoining of DNA ends by NHEJ, resulting in deleterious translocations. 

Genome folding also defines the 3D context in which the DSB response propagates. For example, γH2AX spreading is largely influenced by the folding of chromosomes into TADs, with TAD boundaries correlating with the extent of γ2AX spreading [107]. Furthermore, CTCF-binding sites, which define TAD borders, are enriched around γH2AX foci [108,109]. A functional relationship is further supported by the failure of CTCF-deficient cells to properly assemble γH2AX foci, as well as recent data depicting a role for cohesin in foci formation [108,110]. Importantly, TAD-defined spreading may not be the exclusive mechanism for the propagation of DNA damage response factors, as other proteins, such as 53BP1, can spread over several TADs or sub-TADs in a manner that only partially relies on cohesin [109]. Whether chromosome folding has other functions in repair remains to be investigated. If the pre-existing chromatin architecture is important for the DNA damage response, it is also widely affected in response to DNA damage. Notably, chromatin marks and histone variants are deposited de novo on DSB-flanking sequences, including typical heterochromatin marks. This, along with variations in the chromatin compaction around DSB, plays a central role in DSB repair pathway choice (for a review see [111]). Higher-order chromatin folding is also modified, with the strengthening of TAD boundaries [112,113], an enrichment of TAD cliques and the formation of a new interaction-based subcompartment (D compartment) that groups damaged sequences with nondamaged loci enriched in chromatin marks typical of active transcription (H2AZac, H3K4me3 and H3K79me2; [113]).

These modifications are likely sustained by the increased chromatin dynamics observed in response to DNA damage. Indeed, in *S. cerevisiae*, both the damaged DNA site and the whole undamaged genome increase mobility (Figure 3A; [114,115]). Increased DSB motion has also been observed in *Drosophila* and mammalian cells [88,116]. DNA repair factors, chromatin remodeling complexes and the activity of actin filaments and microtubules have been identified as key elements that facilitate increased DSB chromatin motion (see [117,118] for more details). In budding yeast, decompaction of the chromatin fiber, associated with histone loss, is a key factor in increased chromatin dynamics [119]. This decompaction extends globally, with potential HR donor sequences also becoming more accessible and exploring larger nuclear volumes [80]. Enhanced chromatin movement was first proposed to increase the probability that separated DSB ends find each other prior to NHEJ [120] or to increase the rate of homology search during HDR [114,115]. However, recent work in budding yeast has demonstrated that the mobility of DSB ends was not rate limiting for timely HDR [80]. In this study, the absence of SUMO-dependent ubiquitin ligase Uls1 was shown to compromise local DSB movement, whilst maintaining increased global genome dynamics, DSB resection, checkpoint activation, histone degradation and chromatin decompaction [80]. Despite reduced DSB mobility, homology-directed strand invasion was not delayed, indicating that movement of the break is not limiting for the homology search [80]. Whether global genome mobility is critical for HR efficiency remains to be demonstrated.

Despite this, increased chromatin mobility could facilitate efficient DSB repair in numerous other ways. These include by moving DNA DSBs outside of repair-repressive domains or into domains that favor repair. In line with this, several types of DNA lesions, including DSBs, have been shown to migrate to the nuclear periphery in budding and fission yeast (Figure 3A) and *Drosophila* (Figure 3B; [88,121,122,123,124]). In budding yeast, they associate with two distinct sites, either the nuclear pore complex (NPC) throughout the cell cycle, or the inner nuclear membrane SUN protein, Mps3, in the S/G2 phase. Relocation of DNA lesions to Mps3 or the NPC requires distinct signaling mechanisms, promoting distinct DNA repair pathways (extensively reviewed in [125]). Although relocation of DSBs to the nuclear periphery has not been observed in mammalian cells, displacement of DSBs is nonetheless observed, as demonstrated by DSB relocation outside of heterochromatic domains (Figure 3C; [126]). This is consistent with the need to relocate difficult to repair breaks outside of compartments that are repressive for some repair pathways and to move them towards more favorable environments in which repair could take place. Although a number of studies have described some of the actors required for DSB perinuclear localization, the precise molecular mechanism, from DNA damage to contact with perinuclear anchors, remains to be solved. Similarly, how Mps3 and the NPC define subnuclear compartments favoring repair is still unknown.

## 5. Cohesin in Repair

With the emerging importance of cohesin in shaping the genome by loop extrusion, new aspects of cohesin contribution to DNA damage signaling and repair are appearing.

A role for cohesin in DNA repair was in fact discovered before its well-described role in sister chromatid cohesion, with the *S. pombe* Rad21 gene being identified for providing resistance to ionizing radiation [127]. Since then, cohesin has increasingly been implicated in DNA damage repair, although its function in this was first linked to its capacity to maintain sister chromatid cohesion at the DSB site, to facilitate HDR. Studies in yeast and mammals have demonstrated that cohesin is recruited to DNA DSB sites [107,128,129,130]. How cohesin is enriched and regulated at DSBs remains to be fully described. In yeast and humans, the cohesin loading complexes Scc2/4 and NIPBL–Mau2, respectively, are essential for the enrichment of cohesin at DSBs, suggesting de novo loading is responsible, not rearrangement of preloaded cohesin [128,129,131]. However, the recent finding that NIPBL is required for loop extrusion [16] highlights a possibility for a loop-extrusion-dependent accumulation of preloaded cohesin at DSBs. Strikingly, key components of the DNA damage checkpoint (DDC), the response mechanism that enables the detection and repair of DSBs, are important for cohesin DSB recruitment. MRX^MRN^ and the Tel1^ATM^ kinase are required both in yeast and humans [41,128,131], and γH2AX, the Mec1^ATR^ and Chk1 kinases are also important for cohesin enrichment at DSBs in yeast [128]. Sumoylation of the cohesin subunit Scc1 by the SUMO ligase Mms21 (Mms21/Nse2 in humans) also assists recruitment of cohesin at yeast DSBs [132]. Cohesin binding at DSBs is kept in check by the SUMO-dependent ubiquitin ligase Uls1, whose absence increases MRX and cohesin levels at DSB [80].

The Mms21 SUMO ligase is itself recruited to DSBs by another DSB-binding SMC, the essential Smc5/Smc6 (SMC5/6) complex [133], originally identified in *S. pombe* in genetic screens probing for increased radiation sensitivity [134,135]. Interestingly, SMC5/6 monomers and the holistic complex, including Mms21/Nse2, have been shown to have ssDNA binding affinity, through novel and unique hub and latch domains not found in the other SMC family proteins [136]. Like cohesin, SMC5/6 is enriched in the 25 kb region flanking the DSB [137,138]. Furthermore, knockdown (KD) of the SMC5/6 complex was shown to reduce cohesin loading at DSBs [130]. Crucially, KD of cohesin alone, or together with SMC5/6, resulted in the same reduction in HR events by sister chromatid exchange, indicating that these two complexes act in the same DNA repair pathway [130]. These observations may suggest that the SMC5/6 complex acts as a sensor for DSB ends, leading to the recruitment of the cohesin complex to the DSB. How SMC5/6 senses DSB ends is unknown. One possible hypothesis could be that the ssDNA formed by DSB end resection is detected through the SMC5/6 ssDNA-binding motifs [136]. Another possible mechanism could be linked to the deposition of γH2AX in the DSB-adjacent chromatin. Indeed, Rtt107, a γH2AX-binding protein with which SMC5/6 can interact, is necessary for the enrichment of SMC5/6 at DSBs [139].The full functional role SMC5/6 plays in DNA repair remains unclear, including the mechanism by which it leads to cohesin recruitment. Furthermore, it is possible that SMC5/6 plays roles beyond cohesin recruitment, as demonstrated by the importance of the SUMOylation activity of its Nse2 subunit for the relocation of heterochromatic DSBs in *Drosophila* [90] and DSB interaction with the nuclear periphery in yeast [140]. Whether cohesin is also relevant to these responses remains to be tested.

At DSB sites, local cohesin loading, which is dispensable for sister chromatid cohesion, is key for efficient repair. Indeed, impairing cohesin de novo loading at DSB, in experimental settings that do not affect sister chromatid cohesion, impinges on DNA repair [129,130]. Cohesin has also been proposed to regulate NHEJ in both yeast [141] and human cells [142] through an unknown mechanism. 

More recently, the ability of cohesin to shape individual chromosomes through loop extrusion has been implicated in DNA DSB signaling and repair. One of the first signaling events following DSB induction is the phosphorylation of H2A (H2AX in mammals) by the Tel1^ATM^, Mec1^ATR^ and DNA-PK (only in mammals) PI3-kinases [2,143]. γH2AX can spread over 50–100 kb in yeast [143,144] and over 1–2 Mb of the adjacent chromatin in mammals [145,146] while the kinases appear to be bound close to the DSB ends. Recent studies in human cells have now demonstrated how cohesin-dependent TADs are functional units of the DNA damage response, through γH2AX spreading [131,147]. Hi-C and ChIP-seq data have demonstrated that contacts between the DSB site and distant *cis* chromosome loci are important for establishing γH2AX domains, with the interactome of the break site correlating strongly with the density and spread of γH2AX [147]. These domains are largely defined to TADs, with TAD disruption extending γH2AX spreading into adjacent TADs [147]. Furthermore, DSB sites act as a cohesin translocation roadblock in both yeast and humans [131,148], with cohesin extruding loops away from DSB sites. Therefore, a role for cohesin loop extrusion activity in γH2AX spreading could be imagined, beyond its role in defining TADs with increased interaction [131]. These observations support a model in which cohesin complexes, anchored at DSB ends where the kinase is located, facilitate phosphorylation of H2A as chromatin passes through the cohesin ring during loop extrusion (Figure 4B). In budding yeast, γH2A propagates in both *cis,* and *trans* between nearby genomic regions of different chromosomes [144]; however, the contribution of the cohesin complex and chromosome folding has not been tested.

Loop extrusion by the cohesin complex has also been implicated in the random rearrangement of antibody gene segments of the mouse immune system through a repair process named V(D)J (for a detailed review see [149]). V(D)J recombination is triggered by the programmed formation of DSBs by the RAG endonuclease and results in repair between distant sequences arranged in tandem. Segments destined for rearrangement are interspersed by CTCF sites, which Hi-C data has revealed act as loop anchors and boundaries, limiting contacts and repair between more distant segments [150,151]. Further supporting a functional role for loop extrusion, depletion of cohesin reduces long-range interactions and recombination between distal segments [150], whereas downregulation of WAPL, and thus increasing the size of cohesin-mediated loops, favors repair between more distant segments [152]. Therefore, loop extrusion by cohesin appears to favor intrachromosomal DNA repair between proximal sequences.

Beyond its loop extrusion activity, cohesin may also favor repair with proximal DNA sequences by restricting DSB motion. Indeed, cohesin depletion increases DSB movement beyond the heightened movement observed at DSB sites in WT yeast cells [80]. Accordingly, the interactome around the DSB is altered in absence of cohesin, resulting in increased genome-wide contacts, at the expense of *cis* intrachromosomal interactions (Figure 4C) [148].

Together, these data highlight the contribution of DSB-bound cohesin. Cohesin drives contact between DSB ends and proximal sequences through loop extrusion, participating in DNA damage signaling through γH2AX spreading and promoting intrachromosomal repair. Cohesin also restrains DSB motion, restricting *trans* interactions, further favoring repair with proximal sequences.

Cohesin enrichment is also enhanced genome wide in response to DSB induction [153,154]. In yeast, this enrichment at undamaged sites globally tightens sister chromatid cohesion [30,154]. Similar to the establishment of the S phase cohesion, DSB-induced global cohesin loading relies on Scc2/4, Eco1-mediated Smc3 acetylation and cohesin sumoylation [132,155]. Additionally, DSB-induced phosphorylation of Scc1 by the Chk1 checkpoint kinase is required to allow subsequent Scc1 acetylation by Eco1. Scc1 acetylation counteracts Wpl1 activity, stabilizing cohesin on chromosomes [30,156]. DSB-induced cohesin stabilization may act redundantly with the Chk1-mediated phosphorylation and stabilization of Pds1, antagonizing the activity of the Esp1 separase to delay the metaphase–anaphase transition. In line with this, cohesin accumulates on chromatin upon formation of DNA DSBs [113,130,157] and is involved in the DNA-damage-induced intra-S and G2/M checkpoint activation in human cells [158].

In addition to this, enhanced genome-wide loading of cohesin could mediate the individualization of chromosomes, therefore disfavoring ectopic repair events [148]. Indeed, Hi-C experiments upon HO-induced DSBs in *S. cerevisiae* demonstrated that HR repair occurs in a chromatin context spatially shaped at the global level by cohesin [148]. Whether this relies on pre-existing or de novo loaded cohesin remains to be determined. Cohesin appears to mediate chromosome individualization, reducing overall interchromosomal interactions, which may also restrain the homology search process and promote *cis* dsDNA sampling (Figure 4C) [148]. Accordingly, cohesin depletion increases DSB contacts and favors recombination with the rest of the genome [148]. Importantly, biasing the homology search in *cis* may safeguard the genome against genome instability. 

## 6. Conclusions

While the importance of chromatin composition and organization for DNA repair has become increasingly clear, more work is now required to precisely define the actors and molecular mechanisms at work in these processes. In particular, deciphering how chromatin compaction and the protein or DNA modifications associated with heterochromatin regulate DNA repair pathway choice is crucial, particularly in regard to the development of genome editing tools for therapeutic approaches.

The cohesin complex and its activity as a molecular motor, capable of forming chromatin loops, has emerged as a key player in detecting and responding to DNA damage and, therefore, promoting DNA repair and genome stability. Recent advancements in our knowledge of how this complex works and the technology available for observing its functions at a molecular level, make it likely that we will continue to see novel roles attributed to cohesin for correct DNA repair in the near future. How cohesin interacts with heterochromatin and whether its role there is relevant for DNA repair has not yet been addressed and should be investigated.

A better understanding of cohesin function in DNA repair could be particularly relevant for understanding how cohesin dysfunction affects tumorigenesis. Indeed, cohesin is frequently deregulated in cancer cells, notably in bladder cancer and myeloid neoplasms [159]. The fact that tumors mutated for cohesin have increased sensitivity to DNA damaging agents and PARP inhibitors further suggests a link to their role in DNA repair. Understanding the role of cohesin in DNA repair is, thus, particularly relevant to human health.

## Figures and Tables

**Figure 1 genes-13-00198-f001:**
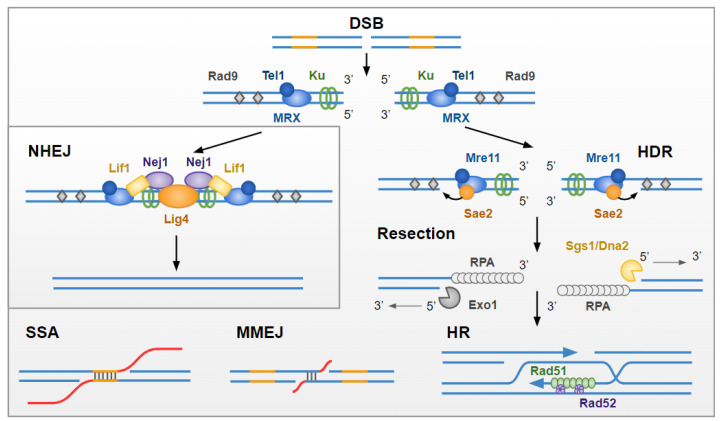
DNA double-strand break repair pathways. DNA double-strand breaks (DSBs) can be repaired by direct re-ligation of broken ends (non-homologous end joining, NHEJ), or through using a homologous template (homology-directed repair, HDR). On the occurrence of a DSB, DNA damage response factors Ku, MRX, Tel1 and Rad9 are recruited to the damaged site. If repair by NHEJ is favored, Lif1, Nej1 and Lig4 are recruited, and broken DNA is re-ligated (see Table 1 for human orthologs). HDR requires the formation of 3′ single-stranded-DNA (ssDNA) overhangs at the DSB site in a process known as resection. Resection is initiated by the endonuclease activity of Mre11 upon stimulation by Sae2 and proceeds due to the activity of the redundant exonucleases Exo1 and Sgs1/Dna2. The 3′ ssDNA overhangs are stabilized by replication protein A (RPA). Rad52 mediates the replacement of RPA for Rad51. Typically, resected Rad51-bound DSB ends undergo repair by homologous recombination (HR), invading the DNA duplex of the replicated sister chromatid for use as a template for faithful DNA DSB repair. Although NHEJ and HR are the canonical DSB repair pathways, other mechanisms are also observed. Repair by microhomology-mediated end joining (MMEJ) is dependent on short ~4–20 bp homologous sequences situated close to the DSB on either side of the break. These short homologous sequences can anneal with one another, sealing the DSB, but generating small deletions (in red). Alternatively, unmasking of longer direct homologous repeats (in orange) can lead to repair by single-strand annealing (SSA), a process that also sees the loss of the genomic sequence that once separated them (in red).

**Figure 2 genes-13-00198-f002:**
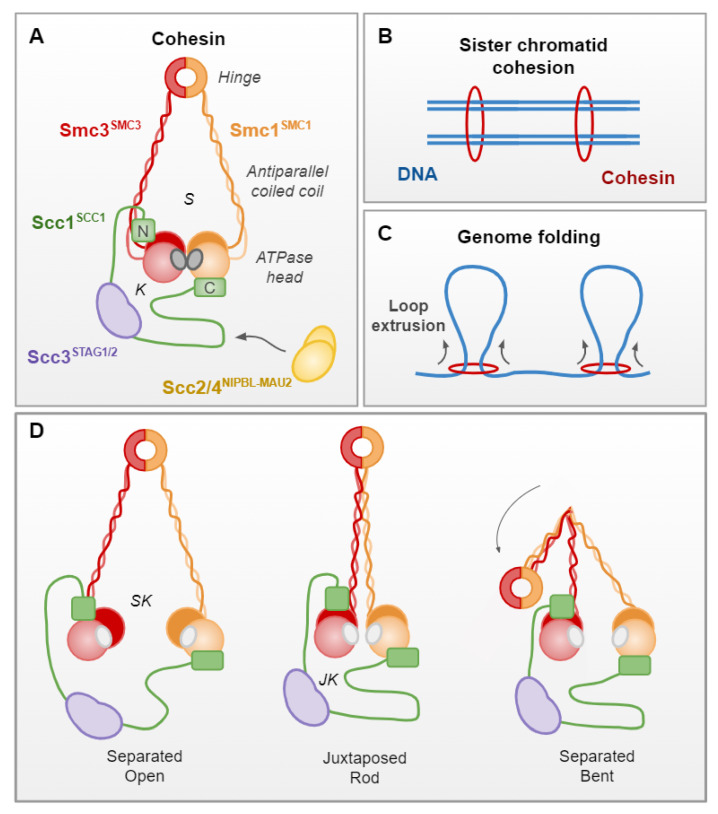
Cohesin structure and molecular functions. (**A**) The cohesin complex, shown in the ATP-bound state, has four core subunits: the structural maintenance of chromosomes proteins, Smc1 and Smc3, the kleisin Scc1 and the kleisin-associating Scc3^STAG1/2^. The loading complex Scc2/4^NIPBLA/NIPBLB–Mau2^ interacts with cohesin through Scc1. SMC proteins consist of ATPase head and hinge domains, and a long antiparallel coiled-coil arm. In the ATP-bound state, closed SMC (S) and kleisin (K) compartments are observed. (**B**) Cohesin holds sister chromatids together from S phase to anaphase. (**C**) Cohesin forms long-range intrachromatid loops, likely by a symmetrical extrusion process. (**D**) Cohesin can exist in multiple conformations determined by ATP binding (SMC heads engaged) and hydrolysis (SMC heads juxtaposed/separated). When separated, the coiled-coil arms generate one open SMC–kleisin (SK) compartment. In the juxtaposed state, the SMC coiled coils align, generating a rod-shaped complex, with a juxtaposed kleisin (JK) compartment. Alignment of the coiled coil is permissive to bending at an elbow region within the arms, bringing the hinge domain into close contact with the SMC3 head domain.

**Figure 3 genes-13-00198-f003:**
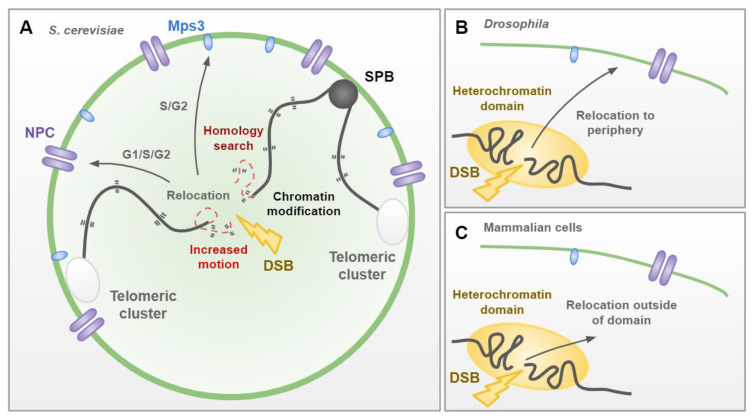
Chromatin dynamics in response to DNA double-strand breaks (DSBs). (**A**) In *S. cerevisiae*, chromosome centromeres are tethered to the spindle pole body (SPB), and telomeres cluster at the nuclear periphery. Upon DSB both local and global processing of the chromatin fiber alter its properties. These chromatin modifications lead to increased chromatin motion of DSB ends and the global genome, which likely assists in the homology search process. Persistent DSBs relocate to the nuclear periphery, through either interaction with the nuclear pore complex (NPC) or Mps3, in a cell-cycle-dependent manner, to assist repair by alternative mechanisms. (**B**) In *Drosophila*, heterochromatic DSBs move out of heterochromatin domains and to the nuclear periphery to facilitate faithful repair. (**C**) In mammalian cells, heterochromatic DSBs move out of heterochromatin domains, but not to the nuclear periphery, to facilitate DSB repair.

**Figure 4 genes-13-00198-f004:**
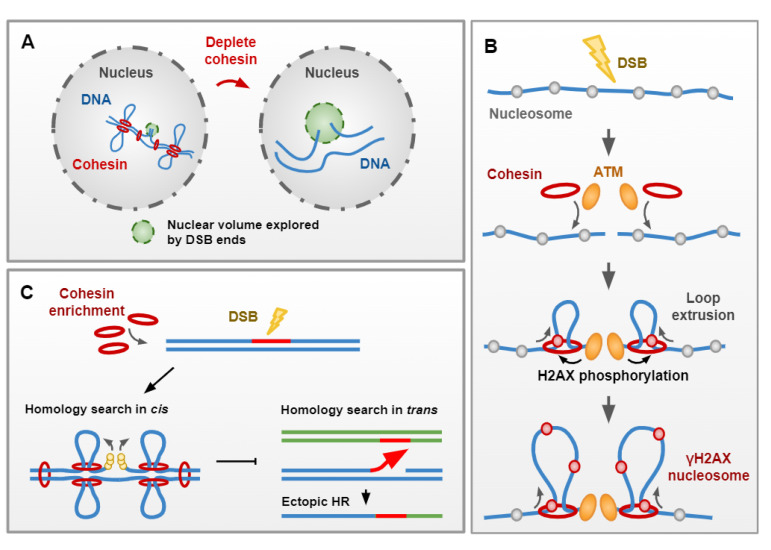
Cohesin contributes to DNA damage signaling and repair. (**A**) Cohesion of sister chromatids restricts chromatin mobility. Cohesin also restricts chromatin mobility in response to DNA damage, with the nuclear volume explored by DSB ends increasing upon cohesin disruption. (**B**) Cohesin-dependent TADs are functional units of the DNA damage response, through γH2Ax spreading. Loop-extrusion activity away from a DSB site drives γH2Ax spreading by the PI3 kinase ATM, allowing the establishment of γH2Ax domains. (**C**) Genome-wide loading of cohesin upon DSB leads to the individualization of chromosomes. Loss of cohesin leads to an increase in interchromosomal interactions and decrease in *cis* dsDNA sampling. Individualization of chromosomes may disfavor ectopic repair events by restraining the homology search process. Preventing interchromosome recombination demonstrates a key role for cohesin in safeguarding the genome against genome instability.

**Table 1 genes-13-00198-t001:** Repair factors in *Saccharomyces. cerevisiae*, functions and orthologs in *Schizosaccharomyces. pombe* and humans.

*S. cerevisiae*	*S. pombe*	Human	Complex	Function
Yku70	Pku70	KU70	KU (DNA-PK)	NHEJ repair factor
Yku80	Pku80	KU80		
-	-	DNA-PKcs		
Lif1	Xrc4	XRCC4	XRCC4–XLF–Ligase IV	
Nej1	Xlf1	NHEJ1 (XLF)		
Dnl4	Lig4	LIG4 (Ligase IV)		
Mre11	Mre11	MRE11	MRX (MRN)	NHEJ/HDR factor
Rad50	Rad50	RAD50		
Xrs2	Nbs1	NBS1		
Sae2	Ctp1	CTIP	-	HDR factor
Dna2	Dna2	DNA2	Dna2/Sgs1 (BLM)	
Sgs1	Rqh1	BLM		
Exo1	Exo1	EXO1	-	
Rad51	Rad51	RAD51	-	
Rad52	Rad52	RAD52	-	
Rad1	Rad16	ERCC4	Rad1–Rad10 (ERCC1–XPF)	
Rad10	Swi10	ERCC1		
Rad9	Crb2	TP53BP1 (53BP1)	-	DNA damage signaling
Tel1	Tel1	ATM	-	
Mec1	Rad3	ATR	-	
Chk1	Chk1	CHEK1	-	

**Table 2 genes-13-00198-t002:** Cohesin subunits in *S. cerevisiae*, functions and orthologs in *S. pombe* and humans.

*S. cerevisiae*	*S. pombe*	Human	Complex	Function
Smc1	Psm1	SMC1A/B	Cohesin	Genome organization Sister chromatid cohesion
Smc3	Psm3	SMC3		
Scc1	Rad21	SCC1		
Scc3	Psc3	STAG1/2		
Scc2	Mis4	NIPBLA/B	Scc2/4 (NIPBL–Mau2)	Cohesin loading partner
Scc4	Ssl3	Mau2		
Pds5	Pds5	PDS5	-	Cohesin regulator
Wpl1	Wpl1	WAPL		
Eco1	Eso1	ESCO1/2		
Esp1	Cut1	ESPL1	-	Cohesin separase
Smc5	Smc5	SMC5	SMC5/6	DNA/chromatin processing
Smc6	Smc6	SMC6		
Mms21	Nse2	NSE2		Ubiquitin ligase

## Data Availability

Not applicable.

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
