# Peer review of "DNA Repair in Space and Time: Safeguarding the Genome with the Cohesin Complex"

_genes, 2022, doi:10.3390/genes13020198_

Round 1
Reviewer 1 Report
This review focused on the involvement of the cohesin complex on genome integrity is very comprehensive, well balanced between studies in the yeast S. cerevisiae model and metazoans, rich in well chosen references and very easy to read figures.
Some comments:
- Part 3: Authors begin by explaining the organization of chromatin from "smaller" to "larger" organization. Thus, they start with chromatin around nucleosomes → Compartments → TADs → Chromosome territories. Since compartments, as described by the authors and in the literature, are larger than TADs, it would be good to restructure Part 3. Start by explaining TADs, then compartments and finally chromosomal territories.
- line 296: Similarly, rearrange from smaller to larger: TADs, TADs-cliques, compartments, chromosome territories . TADs-cliques nomenclature seems rather unusual, the authors should explain it better.
- the first sentence of the abstract : “dealt with robustly” could be replaced by “robustly addressed”
- the conclusion could be a more comprehensive, with more precise openings .
typography:
- line 93: reference
- line 210/126: italics to cerevisiae
- line 370 - The authors may also cite Betts-Lindroos 2006 (https://doi.org/10.1016/j.molcel.2006.05.014).
- line 412: Reference is not in the correct form (Lee:2013ic)
- line 432: Refs
- line 181: post-translational modifications (PTM); PTMs in brackets and not vice-versa.
- revise all references; sometimes pages can be missing i.e. line 486, 501, 533... or the year i.e. line 548
Author Response
We thank the reviewer for their positive assessment of our manuscript
We have improved the manuscript by applying the reviewers suggestions:
- Part 3 and line 296: have been corrected and now start by explaining TADs, then compartments and finally chromosomal territories.
- the first sentence of the abstract : “dealt with robustly” could be replaced by “robustly addressed”. This has been corrected
- The conclusion has been developed and includes more openings
typography:
- line 93: reference This has been corrected
- line 210/126: italics to cerevisiae This has been corrected
- line 370 - The authors may also cite Betts-Lindroos 2006(https://doi.org/10.1016/j.molcel.2006.05.014). This reference has been added.
- line 412: Reference is not in the correct form (Lee:2013ic) This has been corrected
- line 432: Refs The references have been added
- line 181: post-translational modifications (PTM); PTMs in brackets and not vice-versa. This has been corrected
- revise all references; sometimes pages can be missing i.e. line 486, 501, 533... or the year i.e. line 548 The references have been revised
Reviewer 2 Report
Review
genes-1547334
“DNA Repair in Space and Time: Safeguarding the Genome with the Cohesin Complex”
by Jamie Phipps and Karine Dubrana
Dear Editor, Dear Authors,
Thank you very much for your consideration to be involved in the revision process of the review article; “DNA Repair in Space and Time: Safeguarding the Genome with the Cohesin Complex” by Jamie Phipps and Karine Dubrana, submitted to the Genes journal.
In the following review article, the authors summarize the accumulated knowledge on the implication of the Cohesin complex in multiple aspects of genome regulation, including, chromatin composition and dynamics, genome organization, and maintaining sister chromatid cohesion. Moreover, authors have considered the emerging role of the Cohesin complex as a key player in the DNA damage response and its influence on the repair pathways selection.
The review article is very nicely written and most of the key aspects described are supported by the informative illustrations. The review covers a substantial amount of literature, which aggregates in a solid and informative manuscript. However, despite all superlatives, I would like to suggest a few improvements, which could potentially increase the readability of the text and could improve its clarity.
- The manuscript is mainly focused on the role of the Cohesin complex in yeasts, but also includes many examples from higher eukaryotes. This generates a large pallet of proteins, which is difficult to follow. It will be a great improvement if the authors generate a table or multiple tables, including all key players in the described processes and indicated their orthologs/homologs in different species, together with a short description of their functions.
- Figure 3A, should be a separate figure as it belongs to a different chapter from the manuscript.
- Moreover, there is no illustration accompanying the chapter “Genome folding and chromatin dynamics modulate DNA repair”. It is highly advisable if authors are able to generate an illustration summarizing the main concepts described in the chapter.
- Page 10, line 412, the corresponding reference is not properly formatted.
Author Response
We thank the reviewer for their very positive assessment of our manuscript.
Here are the details of the modifications that follow the reviewer suggestions:
- We generated two tables including all key players in the described processes and indicated their orthologs/homologs in budding and fission yeast and humans, together with a short description of their functions.
- We added an illustration to accompany the chapter “Genome folding and chromatin dynamics modulate DNA repair” summarizing the main concepts described in the chapter.
- Page 10, line 412, the corresponding reference is now properly formatted.